# Maternal Diet during Pregnancy Alters the Metabolites in Relation to Metabolic and Neurodegenerative Diseases in Young Adult Offspring

**DOI:** 10.3390/ijms252011046

**Published:** 2024-10-14

**Authors:** Soo-Min Kim, Songjin Oh, Sang Suk Lee, Sunwha Park, Young-Min Hur, AbuZar Ansari, Gain Lee, Man-Jeong Paik, Young-Ah You, Young Ju Kim

**Affiliations:** 1Department of Obstetrics and Gynecology, Ewha Medical Research Institute, College of Medicine, Ewha Womans University Mokdong Hospital, Seoul 07985, Republic of Korea; zeus_0218@ewhain.net (S.-M.K.); clarissa15@gmail.com (S.P.); hym1210@ewha.ac.kr (Y.-M.H.); abu.kim.0313@gmail.com (A.A.); loveleee0102@gmail.com (G.L.); 2Graduate Program in System Health Science and Engineering, Ewha Womans University, Seoul 03760, Republic of Korea; 3College of Pharmacy, Sunchon National University, Suncheon 57922, Republic of Korea; osj7797@naver.com (S.O.); rumen@sunchon.ac.kr (S.S.L.); paik815@scnu.ac.kr (M.-J.P.)

**Keywords:** developmental origins of health and disease (DOHaD), metabolomic profiling, metabolic diseases, neurodegenerative diseases, sex differences

## Abstract

Maternal nutrition during the critical period of pregnancy increases the susceptibility of offspring to the development of diseases later in life. This study aimed to analyze metabolite profiles to investigate the effect of maternal diet during pregnancy on changes in offspring plasma metabolites and to identify correlations with metabolic parameters. Pregnant Sprague-Dawley rats were exposed to under- and overnutrition compared to controls, and their offspring were fed a standard diet after birth. Plasma metabolism was profiled in offspring at 16 weeks of age using liquid chromatography–mass spectrometry (LC-MS/MS) and gas chromatography–tandem mass spectrometry (GC-MS/MS). We analyzed 80 metabolites to identify distinct metabolites and metabolic and neurodegenerative disease-associated metabolites that were sex-differentially altered in each group compared to controls (*p* < 0.05, VIP score > 1.0). Specifically, changes in 3-indolepropionic acid, anthranilic acid, linoleic acid, and arachidonic acid, which are involved in tryptophan and linoleic acid metabolism, were observed in male offspring and correlated with plasma leptin levels in male offspring. Our results suggest that fatty acids involved in tryptophan and linoleic acid metabolism, which are altered by the maternal diet during pregnancy, may lead to an increased risk of metabolic and neurodegenerative diseases in the early life of male offspring.

## 1. Introduction

Maternal undernutrition and overnutrition during pregnancy affect fetal health. Dutch famine birth cohort study revealed that offspring who were malnourished in utero had a greater chance of developing obesity and coronary heart disease in adulthood [1] and that maternal overnutrition was associated with higher birth weight [2] and metabolic disease [3]. In particular, maternal undernutrition and overnutrition alter leptin expression in offspring and may lead to leptin resistance [4,5,6]. In addition, maternal nutrition plays an important role in fetal brain development; low birth weight fetuses born with a disturbed balance between fetal demand and maternal supply have a higher risk of neurodegenerative diseases later in life [7,8]. However, the effects of maternal nutrition during pregnancy on biological changes in offspring and the resulting diseases remain largely unclear.

Our previous studies have shown that, in animal models based on the fetal programming hypothesis, maternal nutritional deficiencies are associated with sex-specific hepatic metabolic dysregulation, altered lipid metabolism, and changes in plasma triglyceride and leptin levels [9,10,11,12]. Prenatal stress and poor nutrition, including inflammation, can have long-term effects on both the mother and child. The microbiome is also altered by prenatal stress and nutrition, which can affect the neurodevelopment, cognition, and behavior of a child [13,14,15]. In addition, the maternal gut microbiome can modulate fetal metabolites, several of which affect neurological conditions [16]. In this context, an urgent need exists for effective approaches to identify risk factors for metabolic and neurodegenerative diseases early in life.

Metabolomes can indicate early biological changes caused by diseases and provide useful biomarkers for identifying early-stage diseases [17]. Alterations in metabolites associated with metabolism may increase the risk of metabolic syndrome, and research on sex-specific changes in the metabolome is needed for disease prevention and treatment in precision medicine. In addition, the metabolome has been used to identify biomarkers in pediatric populations, such as in the study of metabolites associated with insulin resistance in young obese children [18,19,20]. Several studies have identified metabolite profiles that can predict metabolic syndromes, including cardiovascular disease, type 2 diabetes, fatty liver disease, and neurophysiology [19,21,22,23]. These profiles can be used to effectively diagnose and treat these conditions. Therefore, studying the profiles of metabolites associated with disease risk in early life according to sex is essential for predicting diseases and identifying biomarkers.

This study aimed to identify plasma metabolites that are altered by maternal diet during pregnancy and to analyze their correlation with metabolic parameters in a sex-specific manner. We investigated the changes in the metabolites of the offspring compared to those in the controls using GC-MS/MS and LC-MS/MS analyses and explored the correlation between the identified distinct metabolome and metabolic parameters.

## 2. Results

### 2.1. Characteristics and Plasma Metabolic Parameters in Rat Offspring

Based on the maternal diet during pregnancy and lactation, Sprague-Dawley (SD) rats were divided into four groups from day 10 of pregnancy: AdLib/AdLib (control), 50% food restriction/AdLib (FR), 45% high-fat/AdLib (HF), and 45% high-fat/45% high-fat (OB) (Appendix A). At three weeks of age, offspring from the control, FR, and HF groups were fed a standard chow diet, whereas offspring from the OB group were fed a 45% high-fat diet until 16 weeks of age. The food intake of the offspring monitored each week is shown in Appendix A.

After birth, the offspring were weighed at the same time every week until they reached 16 weeks of age. In the offspring at 16 weeks of age, the male FR groups had body weights similar to those of the controls, and the male HF and OB groups had significant increases (*p* < 0.05 and *p* < 0.001, respectively; Table 1). In the female groups, body weight increased only in the OB groups (*p* < 0.001), whereas the other groups were similar. The brain weights of the offspring at 16 weeks of age in the male FR group were significantly lower than those in the controls (Table 1). The adipose tissue weights were significantly higher in the male FR, HF, and OB groups than those in the control group. Glucose levels increased in both the male and female groups, but it was only statistically significant in the male OB group. In the female group, total cholesterol levels significantly increased in the FR group (*p* < 0.05), and in particular, total cholesterol levels increased significantly in the male FR group compared to the male HF group (*p* < 0.05). Plasma triglyceride levels were significantly increased in all-male groups (*p* < 0.05) and in the female FR and OB groups (*p* < 0.05). Furthermore, leptin concentrations were significantly increased in the male FR, HF, and OB groups (*p* < 0.05, *p* < 0.05, and *p* < 0.001, respectively) and in the female OB group (*p* < 0.001). However, glucagon-like peptide-1 (GLP-1) concentrations showed sex differences; males showed increased levels in the FR and OB groups, whereas females showed decreased levels in the HF and OB groups (*p* < 0.05). In addition, there was an increase in the homeostatic model assessment of insulin resistance (HOMA-IR) score in the all-male group, but this was significant only in the male OB group. In the mother, there was a significant increase in the HOMA-IR value in the FR group (Appendix A).

### 2.2. Classification of Significantly Altered Metabolites between Control and Each Groups

Metabolic profiling was conducted using a library database based on standard chemicals. The metabolite levels in the FR, HF, and OB male and female groups were normalized to the corresponding mean values of the control male and female groups, respectively. Star symbol patterns, used for the readily distinguishable discrimination of metabolic alterations, were drawn using normalized values (Appendix A).

Compared to the control and FR male groups, nine metabolites, including arachidonic acid, lysine, and anthranilic acid, were significantly altered (*p* < 0.05). In the FR female group, isoleucine, proline, valine, and homoserine levels were significantly altered compared to those in the control female group (*p* < 0.05). In addition, significant changes were observed in 14 metabolites, including 3-indolepropionic acid, linoleic acid, α-linolenic acid, and asparagine, between the control and HF male groups (*p* < 0.05). Myristic acid, pipecolic acid, and lysine levels were significantly altered in the control and HF female groups (*p* < 0.05). In particular, numerous metabolites were significantly altered in the OB group, with 32 metabolites, including arachidonic acid, lysine, and anthranilic acid, being significantly altered compared to the control and FR male groups (*p* < 0.05). In the OB female groups, 17 metabolites were significantly altered compared with the controls, including 3-indolepropionic acid, lysine, and quinolinic acid (*p* < 0.05).

Partial least-squares discriminant analysis (PLS-DA), a powerful supervised learning tool, was performed on 80 metabolites to evaluate the differential metabolites that greatly contributed to group separation (Figure 1). In addition, receiver operating characteristic (ROC) analysis was used to identify 80 metabolites that had a significant effect in each treatment group compared to the control, considering the *p*-values and VIP scores (*p* < 0.05, VIP score > 1.0, AUC > 0.8, Table 2). Consequently, significant metabolites were confirmed between controls and each group in male (Figure 1A) and female (Figure 1B) offspring, considering *p*-values, VIP score, and AUC (*p* < 0.05, VIP score ≥ 1.5, AUC ≥ 0.8).

PLS-DA completely separated the control group from the FR, HF, and OB groups. Eight metabolites, including arachidonic acid and anthranilic acid, in the male FR group, and 12 metabolites, including linoleic acid and 3-indolepropionic acid, in the male HF group were considered critical metabolites compared to the controls. Eleven metabolites, including 3-indolepropionic acid and arachidonic acid, were considered critical in the male OB groups. In females, PLS-DA completely separated the control group from the FR and OB groups and partially separated the HF group. Isoleucine, caline, homoserine, and proline in the FR group and pipecolic acid, myristic acid, lysine, and creatine in the HF group were considered critical metabolites compared to the controls. In addition, 13 metabolites, including 3-indolepropionic acid and quinolinic acid, were considered critical between the control and OB groups.

### 2.3. Comparison of Metabolites Related to Metabolic Pathways in Offspring Group

In the pathway analysis, pathways with a *p*-value < 0.05 and an impact score > 1.0 in each group were selected (Figure 2). Prominent metabolic pathways were observed in arachidonic acid, tryptophan, and linoleic acid metabolism in the control and FR male groups, as well as in lysine degradation and tryptophan metabolism in the control and FR female groups. In addition, prominent metabolic pathways were identified in linoleic acid metabolism, α-linolenic acid metabolism, cysteine and methionine metabolism, tryptophan metabolism, and lysine degradation between the control and HF male groups, in addition to lysine degradation between the control and HF female groups. Notably, prominent metabolic pathways, including arachidonic acid metabolism, were discovered between the control and OB male groups, and phenylalanine, glycine, serine, and threonine metabolism between the control and OB female groups. Accordingly, we compared the metabolites involved in the metabolism of tryptophan, arachidonic acid, and linoleic acid, which were prominent in the pathway analysis of the male groups compared to that in the controls (Appendix A). Linoleic acid, the precursor of arachidonic acid, was significantly increased in male HF rats, and arachidonic acid was significantly increased in male FR and OB groups. In addition, 3-indolelactic acid and 3-indolepropionic acid (IPA) in the indole pathway were significantly decreased in the male HF and OB groups, whereas anthranilic acid and quinolinic acid in the kynurenine pathway were significantly increased in the male FR and OB groups, respectively.

### 2.4. Correlation with Identified Metabolites Related to Metabolic Pathway in Offspring Group

In the FR male group, leptin was positively correlated with arachidonic acid and anthranilic acid and negatively correlated with lysine and asparagine (Figure 3). In the female FR group, GLP-1 negatively correlated with isoleucine and valine levels. Leptin correlated positively with α-linolenic acid and linoleic acid and negatively with cytidine in the HF male group. In the female HF diet group, GLP-1 positively correlated with myristic acid and negatively correlated with lysine. In the male OB group, leptin was positively correlated with oleic acid, palmitoleic acid, arachidonic acid, and stearic acid, whereas it was negatively correlated with 3-indolepropionic acid and histidine. In the OB female group, leptin was positively correlated with lysine and quinolinic acid and negatively correlated with 3-indolepropionic acid. In particular, GLP-1 levels were negatively correlated with lysine and quinolinic acid levels.

## 3. Discussion

In this study, we found that maternal under- and overnutrition during pregnancy resulted in alterations in plasma metabolites associated with metabolic and neurodegenerative diseases in young adult offspring. Overall, the metabolomic profile analysis showed that the metabolite differences compared to the control group were more pronounced in males. Specifically, metabolites related to the tryptophan pathway and some omega-6 fatty acids involved in linoleic acid metabolism were significantly altered in male offspring compared to controls, whereas female offspring were not significantly altered compared to controls. In the male HF group, 3-indolelactic acid, 3-indolepropionic acid, and linoleic acid were altered, and in the male FR group, arachidonic acid was altered, and plasma leptin, in particular, correlated with omega-6 fatty acids associated with linoleic acid metabolism in male offspring. Our results suggest that alterations in metabolites due to an inappropriate maternal diet during pregnancy may increase the risk of metabolic diseases related to lipid metabolism and neurodegenerative diseases in offspring.

The Developmental Origins of Health and Disease (DOHaD) theory proposes that early life environment determines susceptibility to chronic diseases in adulthood [24,25,26,27]. The recent increase in consuming nutrient-poor processed foods has increased our susceptibility to disease. In particular, the increased consumption of processed foods by mothers during pregnancy is of particular concern, as it may lead to malnutrition in the offspring, further increasing their susceptibility to disease [28,29,30]. A high-fat diet in the mother can alter the microbiome of both the mother and her offspring and the metabolites that are produced by these microbes [31]. In addition, hyperleptinemia in the offspring can be caused by environmental alterations during fetal development due to malnutrition and maternal obesity [13,32]. Our results showed that a significant increase was observed in the plasma level of leptin and accumulation of adipose tissue in male offspring. These results suggest that offspring exposed to maternal malnutrition and overnutrition are more susceptible to metabolic diseases than the offspring of mothers who maintain a normal diet. Additionally, there has been much research on the identification of metabolites in blood or urine that have the potential to be used as biomarkers for the early diagnosis and treatment of a variety of diseases [14,33]. Childhood obesity studies have also identified changes in the plasma metabolome that increase the risk of obesity in adulthood; these metabolomic changes are associated with hyperleptinemia, oxidative stress, hypertriglyceridemia, and inflammation in children with obesity [34,35]. In this study, we found changes in some metabolites associated with metabolic and neurodegenerative diseases in the plasma of young adult offspring exposed to maternal malnutrition and overnutrition.

First, the altered metabolites were associated with the indole pathway, a tryptophan pathway in the male high-fat group. 3-Indolepropionic acid (IPA), which is a gut-derived metabolite resulting from tryptophan metabolism, has been found to have a negative correlation with obesity, type 2 diabetes, and hepatic fibrosis [36,37,38,39]. Another study demonstrated that IPA administration has the potential to improve gut microbiota dysbiosis and attenuate non-alcoholic steatohepatitis (NASH) in rats fed a high-fat diet [40]. In addition to metabolic diseases, neurodegenerative diseases are also associated with IPA. The development of Alzheimer’s disease has neuropathological features that involve the accumulation of toxic proteins in the brain; in particular, IPA has an important role in protecting primary neurons from damage due to the accumulation of amyloid beta protein (Aβ) [41,42].

Another tryptophan pathway, the tryptophan-kynurenine pathway, which has been linked to obesity-related inflammation and metabolic syndrome, differs between childhood and adulthood [43]. Quinolinic acid, a tryptophan metabolite, is produced via the kynurenine pathway and is involved in neurodegenerative diseases [44]. Maternal activation of the immune system and stress may alter levels of kynurenine pathway metabolites in the developing brain [45]. Anthranilic acid is also metabolized by the kynurenine pathway, and its levels are increased in a mouse model of attention deficit hyperactivity disorder (ADHD)/autism spectrum disorder (ASD) [46].

This study found that although offspring were fed a standard diet after birth, IPA concentration was reduced in both the male HF and OB groups. In addition, our results showed that anthranilic acid significantly increased in the male FR group, and quinolinic acid significantly increased in both the male and female OB groups. These findings suggest that maternal under- and overnutrition during pregnancy may increase susceptibility to metabolic and neurodegenerative diseases, particularly in male offspring.

The two major classes of polyunsaturated fatty acids (PUFAs) are omega-3 (n3) and omega-6 (n6). The major omega-3 fatty acid is alpha-linolenic acid, and the major omega-6 fatty acid is linoleic acid. Linoleic acid can also be converted to other fatty acids, such as arachidonic acid, an essential omega-6 fatty acid [47]. Although linoleic acid is an essential fatty acid, its excessive consumption of linoleic acid can lead to increased body weight gain, insulin resistance, and obesity in male rats [48,49]. According to studies in children, fatty acids, including linoleic acid, oleic acid, and palmitic acid, were significantly higher in the obese and NAFLD childhood groups and had a significant positive correlation with triglycerides and cholesterol [50]. Additionally, infants may be at risk of developing obesity if they are exposed to high levels of linoleic acid during adipose tissue development [51]. Arachidonic acid has different effects on males and females and could aggravate obesity by altering the gut microbial composition, especially in male mice, leading to non-alcoholic steatohepatitis (NASH) by increasing total cholesterol and triglyceride levels [47]. Particularly in childhood, elevated levels of arachidonic acid in the adipose tissue have a significant positive association with obesity status [52]. Administration of arachidonic acid reduces hypothalamic leptin sensitivity and attenuates the inhibition of food intake and body weight associated with leptin function in mice [53]. In addition, increased arachidonic acid levels caused by a high-fat diet have been shown to be the first step in the development of inflammation as non-alcoholic fatty liver disease progresses [54]. This study showed that arachidonic acid levels were elevated in the male FR and OB groups, whereas linoleic acid, a precursor of arachidonic acid, was elevated in the male HF group compared to that in the control group. This suggested that the omega-6 fatty acid pathway was activated to a greater extent in the FR and OB male groups. Furthermore, elevated levels of arachidonic acid and linoleic acid were positively correlated with leptin levels. Consequently, maternal under- or overnutrition may alter plasma metabolites, particularly omega-6 fatty acids involved in linoleic acid metabolism in male offspring, and the significant correlation with leptin, suggesting that they may play an important role in the progression of metabolic diseases in early life.

A limitation of our study is that we could not confirm changes in offspring metabolites due to maternal diet during pregnancy when they became adults. However, this study is important because it revealed that metabolites associated with metabolic and neurodegenerative diseases were altered in young adult offspring exposed to maternal under- or overnutrition, even when fed a standard diet after birth. Although the identified metabolites can be considered potential markers for the prediction of disease development in young adults, further research is needed to understand how the metabolome, which plays a role in the development of diseases detected early in life, affects disease development later in adulthood.

## 4. Materials and Methods

### 4.1. Experimental Design

All animal experiments were approved by the Institutional Animal Care and Use Committee (IACUC) of the School of Medicine at Ewha Womans University. Nine-week-old pregnant Sprague-Dawley (SD) rats were purchased from Orient Bio (Orient Bio Inc., Seongnam, Kyunggi-do, Republic of Korea). They were acclimatized for one week in a controlled environment (12/12 h light/dark cycle). Sprague-Dawley (SD) rats were divided into four groups based on their maternal diet during pregnancy and after birth. (I) The control group (control, ad libitum/ad libitum) received a standard diet from the 10th day of pregnancy. To provide a 50% food restriction diet, (II) the food restriction group (FR, 50% food restriction/ad libitum) was fed half the average food intake of the control dams starting on the 10th day of pregnancy. (III) the high-fat group (HF, 45% high-fat/ad libitum) and (IV) the obese group (OB, 45% high-fat/45% high-fat) were fed a 45% high-fat diet. After birth, the control, food-restricted, and high-fat groups were fed a standard chow diet, whereas the obese group was fed a 45% high-fat diet during lactation. The composition of the standard and 45% high-fat diets is shown in Appendix A. Offspring from each group were weighed weekly until they reached 16 weeks of age (n = 8 males and n = 7 females in each group). Sixteen-week-old rat offspring were sacrificed by anesthetization with a mixture of Zoletil (Virbac Taguig, Taguig, Philippine) and Rompun (Bayer, Leverkusen, Germany) administered via intramuscular injection before exsanguination.

### 4.2. Plasma Metabolic Parameter

Plasma metabolic parameters were analyzed in the blood collected via cardiac puncture from offspring at 16 weeks of age. The plasma was separated from a whole blood sample by centrifugation at 3000 rpm for 15 min at 4 °C. Plasma concentrations of glucose, total cholesterol, and triglyceride were measured using an enzymatic colorimetric method with a Cobas 8000 (Roche, Mannheim, Germany). Plasma insulin, leptin, and GLP-1 levels were analyzed using a microplate reader (VersaMax ELISA, Molecular Devices, Sunnyvale, CA, USA) using an Ultrasensitive insulin ELISA Kit (ALPCO, 80-INSRTU-E01), Leptin ELISA Kit (Biovendor, RD291001200R, Brno, Czech Republic), and a rat glucagon-like peptide-1 (GLP-1) ELISA Kit (CUSABIO, CSB-E08117, Houston, TX, USA) according to the manufacturer’s instructions.

### 4.3. Chemicals and Reagents

Standard solutions of all metabolites used for quantitative analysis, including organic acids (OAs), fatty acids (FAs), amino acids (AAs), kynurenic acids (KYNs), and nucleoside (NS), and internal standards (ISs), including 13C2-succinic acid, 3,4-dimethoxybenzoic acid, pentadecanoic acid, lauric-d2-acid, 13C1-phenylalanine, 13C1-leucine, norvaline, and 3-deazauridine, were purchased from Sigma-Aldrich (St. Louis, MO, USA). Methanol (MeOH), HPLC-grade distilled water (DW), acetonitrile (ACN), toluene, dichloromethane (DCM), diethyl ether (DEE), ethyl acetate (EA), and sodium chloride were purchased from Kanto Chemical (Chuo-ku, Tokyo, Japan) and J.T. Baker, Inc. (Phillipsburg, NJ, USA). Sulfuric acid and sodium hydroxide (NaOH) were purchased from Daejung Reagent Chemicals (Siheung, Republic of Korea). Reagents containing O-methoxyamine hydrochloride, trimethylamine (TEA), N-methyl-N-(tert-butyldimethylsilyl) trifluoroacetamide (MTBSTFA), and 1% tert-butyldimethylchlorosilane were purchased from Thermo Scientific (Waltham, MA, USA) and Sigma-Aldrich (St. Louis, MO, USA). All chemicals were of an analytical grade.

### 4.4. Sample Preparation of OA and FA Profile Analysis in Plasma for GC-MS/MS Analysis

OA and FA profiling analyses were conducted using gas chromatography–tandem mass spectrometry (GC-MS/MS) with methoxime (MO)/tert-butyldimethylsilyl (TBDMS) derivatives in the plasma, according to our previous reports. Plasma samples were added to ACN, including ISs (13C2-succinic acid, 3,4-dimethoxybenzoic acid, pentadecanoic acid, and lauric-d2-acid) for deproteinization. After centrifugating at 12,300× *g* for 3 min, the supernatant was moved to DW and adjusted to pH ≥ 12 with 5 M NaOH. To generate MO derivative, O-methoxyamine hydrochloride (1 mg) was added and reacted for 60 min at 60 °C. The aqueous phase was saturated with NaCl and maintained pH 2 using 10% sulfuric acid. Liquid–liquid extraction was sequentially performed using DEE (3 mL) and EA (2 mL). The extracts were completely dried using a gentle stream of nitrogen at 40 °C. MTBSTFA (10 μL) and toluene (20 μL) were subjected to the residue to form TBDMS derivative for 60 min at 60 °C. GCMS-TQ8040, a gas chromatograph mass spectrometer (Shimadzu Corp., Kyoto, Japan) interfaced with a triple quadrupole mass spectrometer, was used for OA and FA profiling analyses in multiple reaction monitoring (MRM) mode for quantitative analysis. An Ultra-2 capillary column (5% phenyl-95% methylpolysiloxane bonded phase; 25 m × 0.20 mm i.d., 0.11 μm film thickness) (Agilent Technologies, Palo Alto, CA, USA) was equipped. The oven temperature of GC was set as follows: 100 °C for 2 min and then increased to 300 °C at a rate of 10 °C/min, with an 8 min holding time. One microliter of the sample was introduced in 10:1 split-injection mode. Helium was used as the carrier gas at a consistent flow rate of 0.5 mL/min. Ar was used as the collision gas. The electron impact ionization mode at 70 eV was used for ionization.

### 4.5. Sample Preparation of AA, KYN, and NS Profile Analyses in Plasma for LC-MS/MS Analysis

AA, KYN, and NS profiling analyses were performed using liquid chromatography–tandem mass spectrometry (LC-MS/MS) in plasma, according to our previous reports. For deproteinization, plasma samples were added to ACN, including ISs (13C1-phenylalanine, 13C1-leucine, and norvaline for AAs; 3,4-dimethoxybenzoic acid for KYN; and 3-deazauridine for NS). After centrifugation at 12,300× *g* for 3 min, the supernatant was centrifuged in a Spin X centrifuge filter tube at 12,300× *g* for 3 min. One microliter of each sample was used for the LC-MS/MS analysis. An LCMS-8050 liquid chromatograph mass spectrometer (Shimadzu Corp., Kyoto, Japan) interfaced with a triple quadrupole mass spectrometer was used for AA, KYN, and NS profiling in the MRM mode for quantitative analysis. An Intrada amino acid column (Imtakt, Portland, OR, USA), 150 mm × 2 mm and particle size of 3 μm, was employed for AA profiling. For KYN profiling, a Zorbax Eclipse XDB-C18 column (Agilent Technologies, Santa Clara, CA, USA), 100 mm × 4.6 mm with a particle size of 3.5 μm, was utilized. Additionally, a Hydro-RP C18 column (Phenomenex, Torrance, CA, USA), 150 mm × 4.6 mm and a particle size of 4 μm, was employed for NS profiling. The nebulizing gas flow rate was configured to 2.0 L/min, whereas the flow rates of the drying and heating gases were adjusted to 10.0 L/min. The interface, desolvation line, and heat block temperatures were set at 300 °C, 250 °C, and 400 °C, respectively. Electrospray ionization mode was used for ionization, and collision-induced dissociation gas was applied at a pressure of 270 kPa.

### 4.6. Star Symbol Pattern Recognition Analysis

The levels of metabolites, including OAs, FAs, AAs, KYNs, and NSs, were determined using their respective standard calibration curves. The mean concentrations of each metabolite in the male and female FR, HF, and OB groups were normalized to the corresponding mean values of the control group. Each normalized value is graphically represented as a line extending from a shared central point. Star symbol plots, designed to facilitate the visual recognition of metabolic alterations, were generated from normalized values using Microsoft Excel 2016 (Microsoft, Redmond, WA, USA).

### 4.7. Statistical Analysis

Data are presented as the mean ± standard deviation (SD), and statistical analysis of characteristics and plasma metabolic parameters was performed using SPSS software ver.18.0.0 (Chicago, IL, USA) by one-way analysis of variance (ANOVA) and Kruskal–Wallis test and as a post hoc Mann–Whitney test. *p*-Value was considered statistically significant with *p* < 0.05. Quantitative data are presented as mean ± standard deviation (SD). Upon transforming the concentration data into log10-transformed values, univariate analysis for statistical comparison was conducted using the Wilcoxon rank-sum test to identify the distinguishing features between the two groups. The comparison results were considered significant at *p*-value < 0.05. Partial least squares discriminant analysis (PLS-DA), hierarchical clustering and correlation heatmap analysis, univariate receiver operating characteristic (ROC) curve analysis, and pathway analysis were performed using MetaboAnalyst 5.0 (https://www.metaboanalyst.ca).

## 5. Conclusions

Our results show that metabolites associated with disease development are altered in young adult offspring, even when fed a standard diet, due to an inappropriate maternal diet during pregnancy. Plasma lipid profiles tended to be higher in the FR group than in the HF group, and, in particular, there was a significant decrease in brain weight and a significant increase in total cholesterol in the male FR group compared with the male HF group. This suggests that maternal-restricted diets may have more adverse long-term metabolic effects than high-fat diets. We also observed male-only changes in fatty acids involved in the metabolism of linoleic acid and arachidonic acid, and metabolite changes correlated more strongly with leptin levels in male offspring, depending on sex. Thus, our findings suggest that maternal undernutrition or overnutrition during pregnancy may increase the susceptibility of male offspring to metabolic and neurodegenerative diseases through alterations in metabolites associated with tryptophan and linoleic acid metabolism.

## Figures and Tables

**Figure 1 ijms-25-11046-f001:**
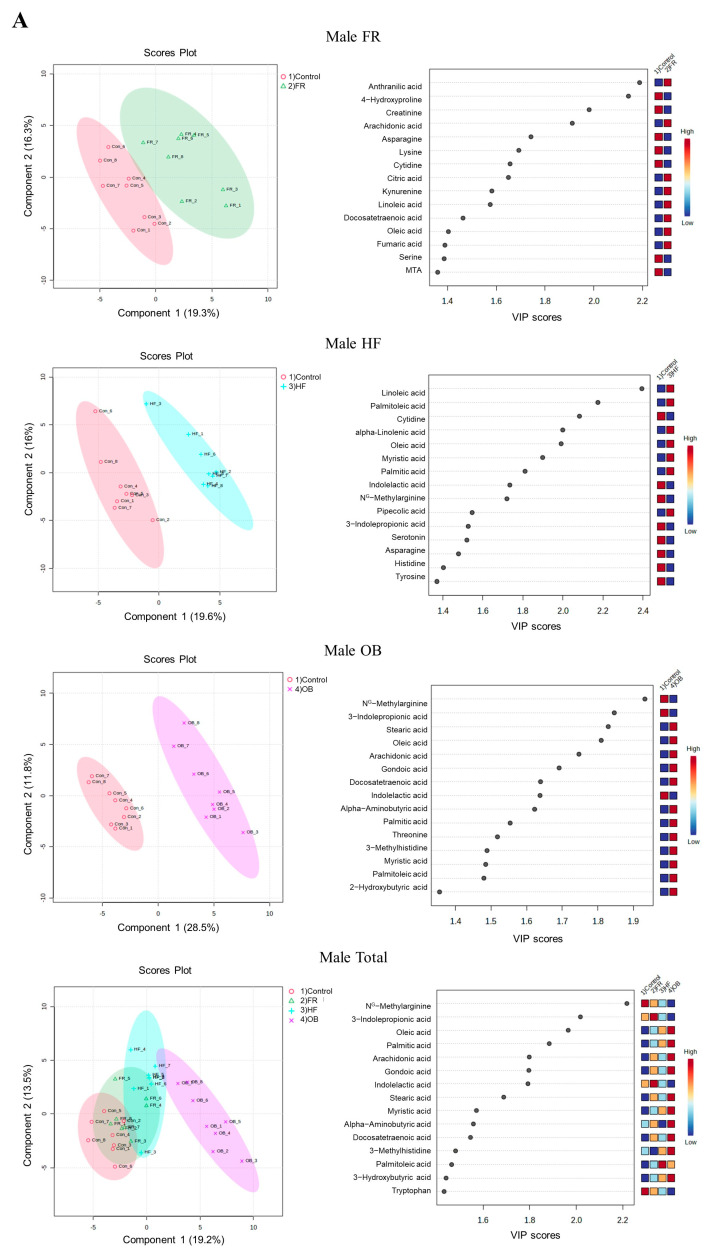
PLS-DA plots showing that metabolites are differentially categorized between male and female groups. Cross-validation and overfitting of PLS-DA were confirmed using parameter accuracy, correlation coefficient (R2), and cross-validation correlation coefficient (Q2). The VIP scores of PLS-DA in the male and female offspring of the FR, HF, and OB groups are presented in Table 2. A Hierarchical clustering heatmap was performed with metabolites (*p* < 0.05), which is a data visualization technique combining hierarchical clustering and grouping similar data points with a heatmap, a graphical representation of data values (Appendix A). Control vs. each group in males (**A**) and in females (**B**).

**Figure 2 ijms-25-11046-f002:**
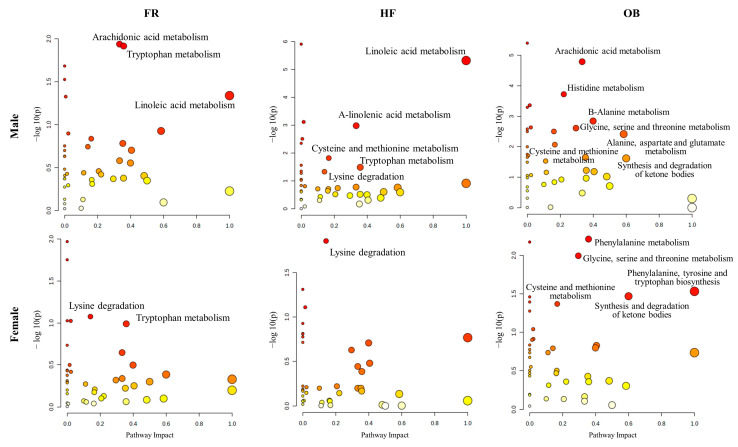
Metabolic pathway analysis of offspring groups. Metabolic pathway analysis was performed using the 80 metabolites, comparing the control group and the FR, HF, and OB groups, respectively. The statistical significance of pathway analysis was indicated by a *p* < 0.05 and an impact score > 0.1.

**Figure 3 ijms-25-11046-f003:**
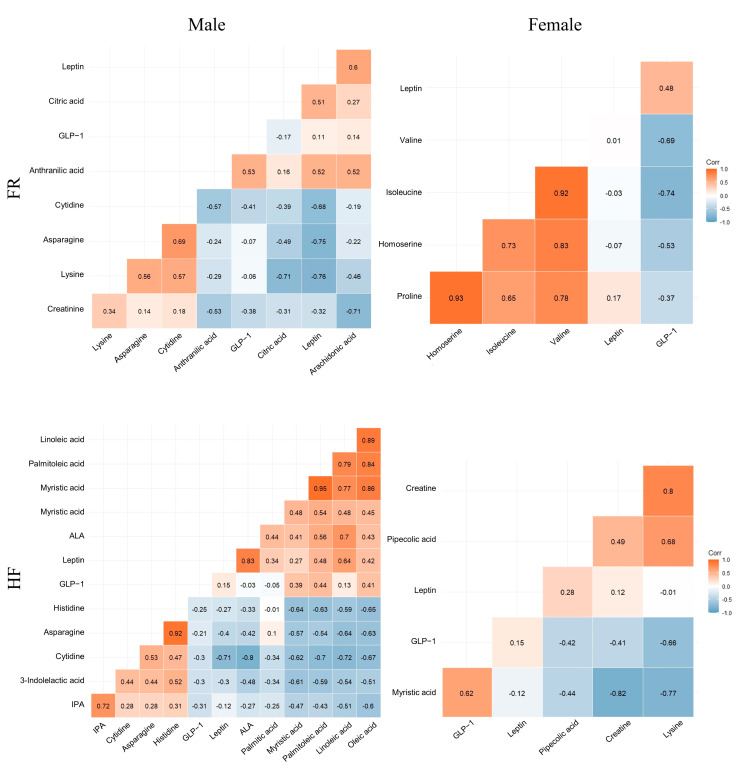
Correlation with metabolites related to metabolic pathway and plasma leptin and GLP-1 levels in the offspring. Hierarchical cluster correlation heatmap analysis and Spearman rank correlation coefficient (R value) were computed to investigate the correlation between the selected metabolites (*p* < 0.05, VIP score > 1.0, AUC > 0.8) and hormones (leptin and GLP-1) in male and female offspring of the FR, HF, and OB groups. The correlation heatmaps are shown in red (positive correlation) and blue (negative correlation). IPA, 3-Indolepropionic acid; 2HB, 2-Hydroxybutyric acid; 3HB, 3-Hydroxybutyric acid; AABA: Alpha-aminobutyric acid; ALA: Alpha-linolenic acid; QA, Quinolinic acid.

**Table 1 ijms-25-11046-t001:** Physiological parameters in male and female rat offspring groups.

Variables	Control (Mean ± SD)	FR (Mean ± SD)	HF (Mean ± SD)	OB (Mean ± SD)	*p*-Value
Male					
Body weight (g)	474.25 ± 33.23	494.84 ± 31.31	519.95 ± 41.57 *	727.20 ± 93.01 **	<0.001
Brain (g)	2.11 ± 0.08	2.00 ± 0.12 *^, †^	2.14 ± 0.08	2.12 ± 0.13	<0.05
Liver (g)	12.90 ± 2.46	14.82 ± 1.05	14.62 ± 1.21	17.21 ± 1.97 *	<0.05
Retroperitoneal adipose tissue (g)	4.75 ± 0.93	8.65 ± 1.69 **	8.90 ± 1.96 **	27.97 ± 11.37 **	<0.001
Glucose (mg/dL)	196.13 ± 15.2	254.0 ± 82.11	201.25 ± 15.83	244.38 ± 44.63 *	<0.05
Total cholesterol (mg/dL)	63.38 ± 5.79	71.25 ± 10.06 ^†^	60.25 ± 5.99	103.00 ± 18.61 **	<0.001
Triglyceride (mg/dL)	51.00 ± 18.93	99.13 ± 35.66 *	89.13 ± 21.04 *	138.25 ± 17.57 **	<0.001
Leptin (ng/mL)	1.78 ± 0.70	4.56 ± 1.53 *	3.54 ± 1.34 *	20.56 ± 8.17 **	<0.001
GLP-1 (ng/mL)	3.74 ± 0.53	4.54 ± 0.85 *	4.16 ± 1.28	5.59 ± 2.4 *	0.074
Insulin (ng/mL)	0.12 ± 0.05	0.14 ± 0.06	0.15 ± 0.05	0.33 ± 0.13 **	<0.05
HOMA-IR	1.48 ± 0.75	1.97 ± 0.59	1.8 ± 0.6	4.97 ± 2.02 **	<0.05
Female					
Body weight (g)	289.09 ± 11.80	288.81 ± 28.06	274.26 ± 25.03	411.96 ± 46.01 **	<0.001
Brain (g)	1.96 ± 0.06	1.87 ± 0.11	1.87 ± 0.12	1.89 ± 0.06 *	0.150
Liver (g)	7.60 ± 0.78	8.27 ± 1.02	7.36 ± 0.62	10.24 ± 3.12 *	<0.05
Retroperitoneal adipose tissue (g)	2.84 ± 0.94	2.97 ± 1.27	3.23 ± 0.99	11.12 ± 3.95 **	<0.05
Glucose (mg/dL)	172.57 ± 23.91	196.43 ± 43.35	196.43 ± 27.67	190.43 ± 17.56	0.343
Total cholesterol (mg/dL)	75.86 ± 7.28	88.86 ± 13.72 *	79.14 ± 8.90	62.71 ± 9.41 *	<0.05
Triglyceride (mg/dL)	27.86 ± 6.40	38.71 ± 9.65 *	32.29 ± 7.74	52.14 ± 14.99 *	<0.05
Leptin (ng/mL)	2.71 ± 1.32	2.54 ± 0.79	2.31 ± 0.75	12.52 ± 6.21 **	<0.05
GLP-1 (ng/mL)	4.26 ± 1.75	3.42 ± 2.30	2.72 ± 0.33 *	2.73 ± 0.52 *	0.091
Insulin (ng/mL)	0.14 ± 0.09	0.08 ± 0.04	0.10 ± 0.06	0.26 ± 0.24	0.081
HOMA-IR	1.52 ± 0.90	1.00 ± 0.52	1.26 ± 0.69	3.18 ± 3.18	0.191

Comparison of physiological parameters, lipid profiles, and hormones among the control, FR, HF, and OB groups (n = 8 males and n = 7 females in each group). Data are expressed as mean ± SD. * *p* < 0.05 vs. control, ** *p* < 0.001 vs. control, ^†^ < 0.05 vs. HF. GLP-1, glucagon-like peptide-1; FR, food restriction; HF, high-fat diet; OB, obese group.

**Table 2 ijms-25-11046-t002:** Levels, *p*-values, VIP scores, and AUC values of differential metabolites from offspring rat models.

Metabolite	Concentration (ng/μL, Mean ± SD)		Normalized Value ^a^	*p*-Value ^b^	VIP Score ^c^	AUC ^d^
Male						
	Control	FR				
Citric acid	6.0 ± 1.5	7.6 ± 1.2	1.27	0.028	1.65	0.83
Arachidonic acid	88.3 ± 11.3	110.2 ± 17.4	1.25	0.010	1.91	0.88
Creatinine	2.1 ± 0.3	1.6 ± 0.2	0.79	0.015	1.98	0.86
4-Hydroxyproline	3.0 ± 0.3	2.4 ± 0.2	0.81	0.0030	2.14	0.92
Asparagine	11.1 ± 2.0	9.0 ± 1.1	0.81	0.021	1.74	0.84
Lysine	144.9 ± 42.6	101.0 ± 29.2	0.70	0.038	1.69	0.81
Anthranilic acid	0.0027 ± 0.0008	0.0044 ± 0.0011	1.62	0.0047	2.19	0.91
Cytidine	0.88 ± 0.12	0.75 ± 0.06	0.85	0.038	1.66	0.81
	Control	HF				
3-Indolepropionic acid	0.47 ± 0.15	0.32 ± 0.07	0.67	0.028	1.53	0.83
3-Indolelactic acid	0.076 ± 0.015	0.052 ± 0.012	0.69	0.0047	1.73	0.91
Myristic acid	1.3 ± 0.4	2.3 ± 0.5	1.79	0.0030	1.90	0.92
Palmitoleic acid	3.7 ± 1.3	9.3 ± 2.2	2.52	0.0011	2.17	0.95
Palmitic acid	73.3 ± 11.1	92.7 ± 9.4	1.26	0.010	1.81	0.88
Linoleic acid	84.2 ± 10.2	126.3 ± 11.4	1.50	0.00016	2.39	1.00
Oleic acid	25.3 ± 6.2	41.9 ± 8.5	1.65	0.0019	1.99	0.94
α-Linolenic acid	2.9 ± 0.9	5.3 ± 1.3	1.84	0.0011	2.00	0.95
Asparagine	11.1 ± 2.0	8.8 ± 1.8	0.79	0.028	1.48	0.83
Histidine	9.4 ± 1.8	7.6 ± 1.4	0.81	0.038	1.40	0.81
*N*^G^-Methylarginine	0.39 ± 0.08	0.27 ± 0.08	0.69	0.010	1.72	0.88
Cytidine	0.88 ± 0.12	0.65 ± 0.06	0.74	0.0047	2.08	0.91
	Control	OB				
2-Hydroxybutyric acid	1.3 ± 0.7	3.8 ± 2.2	2.93	0.0070	1.36	0.89
3-Hydroxybutyric acid	99.5 ± 76.5	296.8 ± 220.7	2.98	0.0070	1.27	0.89
Hippuric acid	0.41 ± 0.10	0.28 ± 0.12	0.66	0.021	1.21	0.84
Malic acid	6.4 ± 0.5	7.2 ± 0.5	1.14	0.015	1.27	0.86
3-Indolepropionic acid	0.47 ± 0.15	0.16 ± 0.01	0.35	0.00016	1.85	1.00
3-Indolelactic acid	0.076 ± 0.015	0.040 ± 0.011	0.53	0.00062	1.64	0.97
Citric acid	6.0 ± 1.5	8.3 ± 1.5	1.39	0.010	1.27	0.88
Myristic acid	1.3 ± 0.4	2.7 ± 0.8	2.04	0.0030	1.48	0.92
Palmitoleic acid	3.7 ± 1.3	7.5 ± 2.0	2.03	0.0011	1.48	0.95
Palmitic acid	73.3 ± 11.1	116.2 ± 22.9	1.59	0.0019	1.55	0.94
Oleic acid	25.3 ± 6.2	65.5 ± 15.5	2.59	0.00016	1.81	1.00
Stearic acid	22.8 ± 5.2	70.5 ± 19.7	3.09	0.00016	1.83	1.00
Arachidonic acid	88.3 ± 11.3	175.7 ± 43.0	1.99	0.00016	1.75	1.00
Gondoic acid	0.46 ± 0.07	0.93 ± 0.25	2.04	0.00031	1.69	0.98
Docosatetraenoic acid	22.8 ± 7.8	61.0 ± 19.7	2.67	0.00062	1.64	0.97
Tryptophan	9.3 ± 1.0	7.3 ± 1.5	0.78	0.007	1.23	0.89
α-Aminobutyric acid	0.83 ± 0.16	2.9 ± 1.6	3.43	0.00016	1.62	1.00
Homoserine	27.5 ± 2.7	35.0 ± 7.0	1.27	0.0070	1.26	0.89
Creatine	56.4 ± 5.9	33.4 ± 21.5	0.59	0.021	1.11	0.84
Threonine	213.4 ± 62.9	379.3 ± 91.5	1.78	0.0019	1.52	0.94
β-Alanine	0.26 ± 0.03	0.43 ± 0.19	1.67	0.0030	1.29	0.92
Asparagine	11.1 ± 2.0	9.0 ± 1.0	0.81	0.021	1.17	0.84
Aspartic acid	0.9 ± 0.3	0.56 ± 0.12	0.61	0.0070	1.28	0.89
Histidine	9.4 ± 1.8	7.4 ± 0.7	0.79	0.0070	1.27	0.89
3-Methylhistidine	1.7 ± 0.2	2.5 ± 0.6	1.53	0.0047	1.49	0.91
*N*^G^-Methylarginine	0.39 ± 0.08	0.12 ± 0.02	0.30	0.00016	1.93	1.00
Quinolinic acid	0.074 ± 0.074	0.13 ± 0.06	1.80	0.015	1.18	0.86
*N*^2^,*N*^2^-Dimethylguanosine	0.0064 ± 0.0006	0.0087 ± 0.0019	1.35	0.010	1.34	0.88
Female						
	Control	FR				
Isoleucine	12.2 ± 1.6	16.4 ± 2.3	1.34	0.0070	2.25	0.92
Proline	15.7 ± 1.4	18.0 ± 2.2	1.15	0.026	1.66	0.86
Valine	23.2 ± 2.9	30.6 ± 4.9	1.32	0.017	2.08	0.88
Homoserine	25.8 ± 2.4	33.1 ± 5.5	1.28	0.0070	2.07	0.92
	Control	HF				
Myristic acid	1.5 ± 0.5	0.9 ± 0.6	0.58	0.038	1.96	0.84
Pipecolic acid	0.37 ± 0.08	0.55 ± 0.12	1.50	0.0041	2.43	0.94
Creatine	37.6 ± 5.0	45.2 ± 6.2	1.20	0.026	1.92	0.86
Lysine	125.5 ± 35.2	179.8 ± 44.6	1.43	0.0070	1.93	0.92
	Control	OB				
2-Hydroxybutyric acid	2.3 ± 1.3	6.5 ± 1.4	2.85	0.0012	1.79	0.98
3-Hydroxybutyric acid	61.8 ± 35.9	169.3 ± 76.8	2.74	0.011	1.61	0.90
2-Hydroxyisovaleric acid	0.27 ± 0.06	0.39 ± 0.07	1.46	0.017	1.54	0.88
Hippuric acid	1.1 ± 0.7	0.23 ± 0.04	0.21	0.00058	1.86	1.00
3-Indolepropionic acid	0.74 ± 0.29	0.18 ± 0.02	0.24	0.00058	2.15	1.00
Docosatetraenoic acid	21.5 ± 5.4	34.5 ± 8.9	1.60	0.011	1.63	0.90
α-Aminobutyric acid	2.0 ± 1.0	4.3 ± 1.2	2.16	0.0041	1.61	0.94
4-Hydroxyproline	1.5 ± 0.2	1.3 ± 0.2	0.85	0.038	1.28	0.84
Homoserine	25.8 ± 2.4	40.7 ± 13.5	1.58	0.017	1.56	0.88
Creatine	37.6 ± 5.0	23.1 ± 8.0	0.61	0.0041	1.71	0.94
Citrulline	55.4 ± 8.7	41.9 ± 10.7	0.76	0.017	1.36	0.88
*N*^G^-Methylarginine	0.26 ± 0.08	0.13 ± 0.03	0.51	0.0023	1.79	0.96
Lysine	125.5 ± 35.2	189.5 ± 49.2	1.51	0.038	1.34	0.84
Quinolinic acid	0.054 ± 0.028	0.11 ± 0.03	2.05	0.0070	1.58	0.92
5-Methylcytidine	0.59 ± 0.05	0.78 ± 0.09	1.33	0.0012	1.91	0.98
*N*^2^,*N*^2^-Dimethylguanosine	0.0064 ± 0.0009	0.012 ± 0.0049	1.80	0.0012	1.58	0.98
1-Methyladenosine	0.0010 ± 0.0001	0.0013 ± 0.0003	1.35	0.026	1.16	0.86

^a^ Values normalized to the corresponding CONTROL concentration values. ^b^
*p*-Value evaluated by Wilcoxon rank-sum test. ^c^ Variable importance in projection score of PLS-DA. ^d^ Values of univariate area under the receiver operating characteristic curve.

## Data Availability

All data generated or analyzed during this study are included in this published article and its Appendix A or are available from the corresponding author upon reasonable request.

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
