# Peer review of "Maternal Diet during Pregnancy Alters the Metabolites in Relation to Metabolic and Neurodegenerative Diseases in Young Adult Offspring"

_ijms, 2024, doi:10.3390/ijms252011046_

Round 1

Reviewer 1 Report

Comments and Suggestions for Authors

Please check the language usage in the manuscript, such as the article's title.

In the abstract, please be more specific and indicate the molecules that change levels in more detail.

Could you provide any information on the pregnant rats (glucose, insulin levels in serum for example)?  Also, this data is missing in the offprints.

Also, discuss the meaning of using different diets during pregnancy and lactation (what is the physiological meaning compared with humans?). This strategy makes it difficult to interpret the data: For example, the lactation period attending to body weight is more relevant than the prenatal period of feeding.

The GLP-1 levels show a different evolution in male and female offprints. Furthermore, the parameter has no clear evolution from restricted to control, HF, and obese male or female animals. Could you provide data on the animals' food intake (in kcal)?

The comparisons between the control and one treatment are mainly presented. It would be advisable to present multiple comparisons among all groups to get a clear picture.

Comments on the Quality of English Language

Please check the manuscript.

Author Response

Responses to editor and reviewer comments

5 October, 2024

We are sincerely grateful for your thorough consideration and scrutiny of our manuscript, “Maternal diet during pregnancy alters the metabolites in relation to metabolic and neurodegenerative diseases in young adult offspring”. We have revised the manuscript according to the Reviewer’s suggestions. The changes within the revised manuscript were highlighted (yellow). Point-by-point responses to the reviewers’ comments are provided below.

[Reviewer 1]

Comment 1: Please check the language usage in the manuscript, such as the article's title. In the abstract, please be more specific and indicate the molecules that change levels in more detail.

Response 1: Thank you for pointing this out. We have changed this according to your suggestion. Also, we have indicated the specific changed molecules in abstract according to your suggestion. [Page, 1; line, 26]

Comment 2: Could you provide any information on the pregnant rats (glucose, insulin levels in serum for example)?  Also, this data is missing in the offprints.

Response 2: Following your suggestions, we have shown the plasma carbohydrate and lipid profiles mother in each group. In addition, we have shown that glucose, insulin, HOMA-IR of the offspring in Table 1. [Page, 3; line, 100]

Comment 3: Also, discuss the meaning of using different diets during pregnancy and lactation (what is the physiological meaning compared with humans?). This strategy makes it difficult to interpret the data: For example, the lactation period attending to body weight is more relevant than the prenatal period of feeding.

Response 3: The theory of developmental origins of health and disease, or DOHaD, proposes that the early life environment in the womb determines susceptibility to chronic disease in adulthood. One of the most important risk factors in these processes is developmental malnutrition. According to the Dutch Famine Birth Cohort Study, prenatal exposure to malnutrition was associated with lower birth weight. In addition, catch-up growth, where babies born with a low birth weight reach or exceed normal body weight later in life, has been shown to be associated with a high risk of disease later in life. Therefore, it's important to eat a healthy, balanced diet during pregnancy to give your baby all the nutrients it needs to grow and develop. In this study, the maternal dietary environment during pregnancy consisted of a restricted diet and a high-fat diet to determine the effects of the maternal diet on the fetus during fetal development.

Comment 4: The GLP-1 levels show a different evolution in male and female offprints. Furthermore, the parameter has no clear evolution from restricted to control, HF, and obese male or female animals. Could you provide data on the animals' food intake (in kcal)?

Response 4: In our study, we found that GLP-1 levels differed between the male and female groups, but more research is needed to determine exactly what mechanisms underlie these differences. We have presented the dietary intake of the offspring in Supplementary Figure 2. Similar food intake was observed in the control, FR and HF groups in males and females, respectively. [Page, 2; line, 81]

Comment 5: The comparisons between the control and one treatment are mainly presented. It would be advisable to present multiple comparisons among all groups to get a clear picture.

Response 5: Following your suggestions, we have therefore provided the p-values from the analysis of the Kruskal-Wallis test to indicate that there were significant differences between the groups in Table 1. [Page, 3; line, 104]

Reviewer 2 Report

Comments and Suggestions for Authors

The study must be reproducible. In this case you have to detail the diets that you use. What means ,,food restriction''? Are you referring to the quantity of food or you changed something in the diet composition? Also, what means ,,high-fat diet''? You add some new components into the diets? which onw?Taking  into account that you said that you had restricted food and high-fat food you have to present the chemical composition of each type of diet and to highlight the differences between the group's diets.

All the figures are too small and hard to be read. I recommend to rethink the figures that you want to present in the paper, to choose only few figures, try to do some charts that represent the most important values that you want to highlight and, maybe, to replace some figures with tables that can be more understandable.

Please try to highlight, in Conclusions, the differences you observed in FR and HF groups, to be easily to understand the impact of undernutrition or overnutrition on the offspring's health.

Please add some references from 2024.

It seems that the present study represents a part of a wider research of the authors, on the presented subject. Which shows an increased interest of the collective for this topic and makes it of interes for the scientific comunity.

Author Response

Responses to editor and reviewer comments

5 October, 2024

We are sincerely grateful for your thorough consideration and scrutiny of our manuscript, “Maternal diet during pregnancy alters the metabolites in relation to metabolic and neurodegenerative diseases in young adult offspring”. We have revised the manuscript according to the Reviewer’s suggestions. The changes within the revised manuscript were highlighted (yellow). Point-by-point responses to the reviewers’ comments are provided below.

[Reviewer 2]

Comment 1: The study must be reproducible. In this case you have to detail the diets that you use. What means ,,food restriction''? Are you referring to the quantity of food or you changed something in the diet composition? Also, what means ,,high-fat diet''? You add some new components into the diets? which onw?Taking  into account that you said that you had restricted food and high-fat food you have to present the chemical composition of each type of diet and to highlight the differences between the group's diets.

Response 1: We think it's a very good point, and we appreciate the reviewer's comment. We have added information on the standard and high-fat diets used in Supplementary Table 4. The standard diet was purchased with a composition of 59% carbohydrate, 27% protein and 14% fat (Altromin, 1314) and the 45% high fat diet was purchased with a composition of 35% carbohydrate, 20% protein and 45% fat (Research Diets, D12451). In the case of food restriction, we measured the average feed intake of the control group and each dam in the FR group of similar body weight to the control group was given 50% of the feed intake of the control group from day 10 of pregnancy until delivery. [Page, 15; line, 335]

Comment 2: All the figures are too small and hard to be read. I recommend to rethink the figures that you want to present in the paper, to choose only few figures, try to do some charts that represent the most important values that you want to highlight and, maybe, to replace some figures with tables that can be more understandable.

Response 2: We have changed this according to your suggestion. We have reduced the number of figures in the manuscript and have shown them in the supplementary data.

Comment 3: Please try to highlight, in Conclusions, the differences you observed in FR and HF groups, to be easily to understand the impact of undernutrition or overnutrition on the offspring's health.

Response 3: We have explained that there was a significant decrease in brain weight and a significant increase in total cholesterol in the male FR group compared with the male HF group. Although not significant, we also observed that plasma metabolic parameters including glucose, total cholesterol, triglycerides and leptin tended to be higher in FR than in HF in both male and female offspring. This suggests that a restricted maternal diet may have more adverse metabolic effects in the long term than a high-fat diet. [Page, 16; line, 448]

Comment 4: Please add some references from 2024.

Response 4: We have added references published in 2024. [Page, 19; line, 537]

Round 2

Reviewer 1 Report

Comments and Suggestions for Authors

The new version of the manuscript, and the data provided, improve its quality and support the main conclusions